# OpenReview forum: "Differentially Private Steering for Large Language Model Alignment"
_ICLR.cc/2025/Conference — ICLR 2025 Poster_

### Official Review · Reviewer_Hvwa · 2024-11-02

**Soundness:** 4
**Presentation:** 4
**Contribution:** 3
**Rating:** 8
**Confidence:** 3

**Summary:**

The paper introduces a novel algorithm called Private Steering for LLM Alignment (PSA) to edit LLM activations with differential privacy (DP) for aligning LLMs. The paper provides background on a technique for "steering" LLM inference by editing the activations of a subset of layers. Usually the activations are edited using a learned vector per layer which are added to the activations of the corresponding layer. Usually this vector is learned from a dataset containing paired examples of the form (prompt, positive completion, negative completion). One method to learn this vector called Mean Steering is to take the mean of the difference of the activations on each layer for all the pairs in the dataset. The authors propose PSA which clips the contribution from each paired example and adds calibrated gaussian noise to obtained a private version of the learned vector using the Gaussian mechanism. This private vector per layer can then be used during inference as needed while protecting the privacy of the examples in the alignment dataset. The privacy of the inference can be somewhat controlled by choosing how many layers should the activation editing be applied to.

The authors also conduct experiments to show the effectiveness of this algorithm compared to Mean Steering and another algorithm called PCA Steering. They align the model using a variety of datasets released as part of Anthropic's "Advanced AI Risk" evaluation. They measure alignment using previously defined benchmarks such multiple choice questions to choose correct behavior and using GPT-4 to rate open-ended text generation. The authors also measure general purpose text generation performance using GPT-4 for open ended text generation and MMLU. The study find that PSA performs similar to Mean Steering and better than PCA Steering for both alignment and general quality and it even outperforms the non-private versions on some benchmarks.

Finally, the study also evaluates empirical privacy of this technique. To test this, the authors propose a novel algorithm which adds a canary containing a prompt token and one of two completion token to the alignment dataset. During inference, the authors add the prompt token in the query and check whether the first completion token is present in the response. They measure the performance of this classifier to obtain an empirical epsilon for the algorithm and find that the algorithm is much more private empirically than the theoretical privacy guarantee.

**Strengths:**

- The paper introduces a novel algorithm for protecting the privacy of the alignment examples while using it for activation editing. Alignment in LLMs is a very important problem and alignment datasets are usually small and could easily leak any private information that is contained in them.
- The paper is easy to follow and the algorithm and experiments are well explained.
-

**Weaknesses:**

- The algorithm is a very simple application of the Gaussian mechanism
- It's not clear how significant the experiments are as there are no error bars. For some of the evaluations, it is not clear how many examples were used.

**Questions:**

- Is it expected that alignment can improve the performance on MMLU? Many of the alignment numbers seem to be better than Zero-shot.
- For some of the datasets, it seems like the delta would actually be fairly large?

---

> ### Author Response · Authors · 2024-11-20
> **Official Response by Authors**
>
> We thank the reviewer for their careful review of the paper, their comments, and their positive assessment of our work. We agree with the reviewer that protecting the privacy of the very small alignment datasets is an emerging problem, and we hope our simple but performant algorithm is a useful solution to this problem.
>
> > Q. **“simple application of Gaussian mechanism”**
>
> A. We believe the relative simplicity of our contribution is a strength. We observe strong results across four model families and seven benchmarks, motivating our approach as a reliable and simple solution which has not been explored in literature before. Additionally, our approach is easy to implement and more likely to be adopted in practical uses because of its simplicity.
>
> > Q. **“bar plot and number of examples used”**
>
> A. We thank the reviewer for pointing this out and will update the plots in the draft to reflect the error bars.  We have provided the standard deviation for our results with the Llama model family below:
>
> |  | Sycophancy | Hallucination | Refusal | Survival Instinct | Myopic Reward | AI Coordination | Corrigibility |
> |---|---|---|---|---|---|---|---|
> | Mean Steer | 66.7 $\pm$ 0.1 | 84.3 $\pm$ 0.02  | 73.7 $\pm$ 0.2 | 47.7 $\pm$ 0.5 | 82.3 $\pm$ 0.8 | 34.3 $\pm$ 0.2 | 89.4 $\pm$ 0.1 |
> | PSA | 65.8 $\pm$ 0.05 | 83.4 $\pm$ 0.1 | 74.6 $\pm$ 0.09 | 48.3 $\pm$ 0.1 | 80.1 $\pm$ 0.7 | 34.1 $\pm$ 0.1 | 89.1 $\pm$ 0.06 |
>
> We will need some additional time to also run these experiments for the other model families but we expect those results to be similar. We will update the revised draft with those numbers. The number of examples in each dataset are reported in Table 2.
>
> > Q. **“Is it expected that alignment can improve performance on MMLU?”**
>
> A.  Investigating the reasoning performance of aligned LLMs is a significant area of contemporary research [1,5]. Our results are in line with similar works [1,3] that show reasoning gains with aligned LLMs. (See Table 5 in [3]). Wang et. al. [1] discover that alignment is crucial for improving reasoning performance of LLMs on math and commonsense datasets while Luo et. al. [2] show that using preference (alignment) data improves the mathematical competence of LLMs.
>
> > Q. **“For some of the datasets, it seems like delta would be fairly large?”**
>
> A. The reviewer is correct in pointing out that $\delta$ can be large when the dataset size $n$ is too small. We are guided by the principle that the parameter $\delta$ is typically set to be less than $1/n$ to ensure meaningful privacy guarantee. This can be explained by the “name and shame” algorithm [4], which outputs each data point with probability $\delta$. Although this algorithm is $(0, \delta)$-DP, the probability of releasing at least one data point is $1-(1-\delta)^n\approx \delta n$. When $\delta \gg 1/n$, this guarantees at least one data point will be released, making the privacy guarantee meaningless. Therefore, we set $\delta = 1/5n$ instead of a fixed value to account for the varying sizes of different datasets. In our experiments, the smallest dataset size is 290 (Table 2), resulting in a sufficiently small $\delta \leq 0.0007$. For such small datasets, making the delta even smaller leads to unreasonably large loss in utility.
>
> [1] Wang, Peiyi, et al. "Making large language models better reasoners with alignment." arXiv preprint arXiv:2309.02144 (2023).
>
> [2] Luo, Haipeng, et al. "Wizardmath: Empowering mathematical reasoning for large language models via reinforced evol-instruct." arXiv preprint arXiv:2308.09583 (2023).
>
> [3] Panickssery, Nina, et al. "Steering llama 2 via contrastive activation addition." ACL 2024
>
> [4] Smith, A. “Lecture note of BU CS 591 Algorithms in Society: lecture 9 and 10. ” 2020.
>
> [5] Jiang, Ruili, et al. "A Survey on Human Preference Learning for Large Language Models." arXiv preprint arXiv:2406.11191 (2024).

---

> > ### Comment · Reviewer_Hvwa · 2024-12-02
> >
> > Thank you for the responses. I will keep my score.

---

### Official Review · Reviewer_DS5g · 2024-11-03

**Soundness:** 3
**Presentation:** 3
**Contribution:** 3
**Rating:** 6
**Confidence:** 4

**Summary:**

The paper addresses the challenge of aligning Large Language Models (LLMs) with human values and reducing undesirable behaviors, like hallucinations, by using activation editing—a method that adjusts model activations based on examples of desired and undesired behaviors. When these examples come from private datasets, there's a risk of leaking sensitive information. To address this, the authors introduce the Private Steering for LLM Alignment (PSA) algorithm, which ensures that activation editing respects differential privacy (DP).

**Strengths:**

S1: Activation editing techniques are appealing because they provide a lightweight alternative to costly iterative optimization, making it easier to adjust LLM behavior efficiently. Consequently, activation editing is emerging as a practical substitute for resource-intensive fine-tuning. However, concerns around potential data leakage risks introduced by these techniques are new, and the solution offered through differential privacy, presents an interesting approach to address these challenges.

S2: Evaluation on seven benchmarks with various open-source LLMs, showing strong alignment with minimal performance loss.

S3: A new Membership Inference Attack (MIA) was designed to empirically evaluate the privacy of LLMs undergoing activation editing.

**Weaknesses:**

Important details are missing (please refer to the following section: 'Questions').

**Questions:**

Q1: In Algorithm 2, $\sigma=\frac{2 \sqrt{2 \ln(1.25/\delta)}}{n\varepsilon}$, which implies that $\Delta_f=n/2$. Does this mean that $C_l$ is equal to $\Delta_f$, and thus equal to $n/2$?

Q2: What is the value of $\varepsilon$ per layer in $\sigma$?

---

> ### Author Response · Authors · 2024-11-20
> **Official Response by Authors**
>
> We thank the reviewer for their careful review and the two sets of specific questions. We are also encouraged by the scores given by the reviewer on Soundness, Presentation, and Contribution. We hope the answers below along with the new experimental data satisfactorily answers the reviewer’s queries and they will consider increasing their scores.
>
> > Answer to Q1 - **clarification on the sensitivity of our algorithm**
>
> A. We thank the reviewer for raising this important question. In algorithm 1, $C_l$ is an input parameter to the algorithm, and the sensitivity of the Gaussian mechanism is $\Delta_f = 2/n$. We clarify as follows.
>
> We note that $C_l$ serves as an estimated upper bound for the norm of difference vectors, similar to the clipping parameter in DP-SGD [1]. Our approach first clips the difference vector to $C_l$ and then divides this clipped vector by $C_l$, ensuring the resulting vectors have norms less than 1. Hence, the sensitivity $\Delta_f$ for mean estimation of these vectors is upper bounded by $2/n$, since changing a single element will only affect the output by at most 2, given that the norm of $\bar{d}_i^l$ is constrained to be less than 1.
>
> Additionally, there is a typo on line 227 in algorithm 1. The equation should read $\bar{d}_i^l = d_i^l / \max\{C_l, ||d_i^l||_2\}$. This equation can be obtained by combining the clipping and normalization steps described above: first, clip the difference vector to $C_l$, which can be done by $d_i^l = d_i^l / \max(1, \frac{||d_i^l||_2}{C_l})$; then, divide the clipped difference vector by $C_l$ to ensure that $\bar{d}_i^l$ has a norm smaller than 1.
>
> > Answer to Q2 - **per layer $\epsilon$ in $\sigma$:**
>
> A. The per-layer $\epsilon$ is $\frac{2\sqrt{2\ln 1.25/\delta}}{kn\sigma}$, where $\delta$ represents the overall $\delta$.
> As described in lines 258–261 of our paper, we add the same amount of noise to each layer, ensuring that the activation editing on each layer is $(\epsilon/k, \delta/k)$-DP. Since we edit a total of $k$ layers, the composition properties of differential privacy imply that the overall process is $(\epsilon, \delta)$-DP. We use basic composition instead of advanced composition of differential privacy, as basic composition provides tighter bounds when $k$ is small.
>
> In our experiments, $k = 5$ as we apply activation editing on 5 middle layers. The values are summarized in the table below.
>
> | Attribute            | Sycophancy | Hallucination | Refusal | Survival Instinct | Myopic Reward | AI Coordination | Corrigibility |
> |----------------------|------------|---------------|---------|-------------------|---------------|-----------------|---------------|
> | Per-layer $\epsilon$ | 0.4        | 0.4           | 0.94    | 0.46              | 0.42          | 1.08            | 1.32          |
> | Total $\epsilon$     | 2.0        | 2.0           | 4.7     | 2.3               | 2.1           | 5.4             | 6.6           |

---

> > ### Comment · Reviewer_DS5g · 2024-11-24
> >
> > First of all, I would like to thank you for taking the time to address my questions.
> >
> > **The per-layer $\epsilon$ is $\frac{2\sqrt{2\ln 1.25/\delta}}{kn\sigma}$, where $\delta$ represents the overall $\delta$.**
> >
> > There is a fundamental issue with the returned values. In order for the formula $\frac{2\sqrt{2\ln 1.25/\delta}}{kn\sigma}$ to be valid, epsilon must be between 0 and 1, as stated in Theorem 3.22 of [1]. However, some cases in your table show per-layer epsilon values greater than 1, which consequently invalidates the privacy guarantees you aim to provide through differential privacy.
> >
> > Additionally, could you please clarify the notion of neighboring datasets that you are considering in your paper?
> >
> > [1]: Dwork, Cynthia, and Aaron Roth. "The algorithmic foundations of differential privacy." Foundations and Trends® in Theoretical Computer Science 9.3–4 (2014): 211-407.

---

> > > ### Author Response · Authors · 2024-11-25
> > >
> > > > Q1.  **“epsilon must be between 0 and 1”**
> > >
> > > We thank the reviewer for raising this interesting point. The reviewer is indeed right that the analysis in Theorem 3.22 from [1] only applies for $\epsilon< 1$. A tighter analysis that works for all $\epsilon$ is provided in Theorem 9 in  [2]. Their work identifies that the original privacy analysis of the Gaussian Mechanism (Theorem 3.22 in [1]) breaks down in the high-$\epsilon$ regime due to errors introduced by tail approximations in the proof. However, **this gap becomes impactful only for very large $\epsilon$ unlike our use case where $\epsilon$ is very close to $1$. Nevertheless, [2] directly calibrates the noise added by the Gaussian Mechanism using the Gaussian cumulative density function, allowing a tighter privacy analysis in such cases.**
> > >
> > > We applied this analysis on the two smaller datasets (Corrigibility and AI Coordination) where per-layer epsilon is 1.08 and 1.32, using their publicly available code. **We evaluated the $\sigma$ values required to achieve the claimed per-layer $\epsilon$. The values are respectively 0.0143 and 0.0147, which are both below 0.02 which we use in our experiments.** We will add this clarification in the updated draft.
> > >
> > > The reviewer is welcome to try out the code at https://raw.githubusercontent.com/BorjaBalle/analytic-gaussian-mechanism/refs/heads/master/agm-example.py
> > > with
> > > ```python
> > > calibrateAnalyticGaussianMechanism(epsilon=1.32, delta=1.0/(5*290), GS=2.0/290, tol = 1.e-12)
> > > ```
> > > and
> > > ```python
> > > calibrateAnalyticGaussianMechanism(epsilon=1.08, delta=1.0/(5*360), GS=2.0/360, tol = 1.e-12)
> > > ```
> > >
> > > > Q2. **“notion of neighbouring datasets in our case”**
> > >
> > > We consider neighbouring datasets $D_\mathrm{priv}$ and $D^{'}_\mathrm{priv}$ that differ in one demonstration. This is the substitution model of neighbouring datasets commonly considered in DP literature. Section 3 describes the data setup in more detail and Table 1 shows an example demonstration.
> > >
> > >
> > > We hope our responses adequately address your concerns and you will consider raising the score. Please let us know if there are more questions, we are happy to discuss!
> > >
> > > [1] Cynthia Dwork, Aaron Roth. “Algorithmic Foundation of Differential Privacy. ” 2014.
> > >
> > > [2] Borja Balle, Yu-Xiang Wang. “Improving the Gaussian Mechanism for Differential Privacy: Analytical Calibration and Optimal Denoising. ” ICML. 2018.

---

> > > > ### Comment · Reviewer_DS5g · 2024-11-27
> > > >
> > > > Thank you for addressing my concerns in the rebuttal. Everything is now well-resolved, and I’ve updated my score accordingly!

---

### Official Review · Reviewer_L9b9 · 2024-11-04

**Soundness:** 3
**Presentation:** 4
**Contribution:** 2
**Rating:** 6
**Confidence:** 4

**Summary:**

This paper investigates the topic of differentially private (DP) inference-time alignment for large language models (LLMs). The proposed approach builds on existing "linear steering vector" methods, which use labeled data to compute a directional vector by contrasting mean representations of positive and negative samples, then add the directional vector to the test-time representation to steer outputs toward the desired alignment objective. The paper establishes a DP guarantee for this alignment process through DP mean computation. Experimental results demonstrate that this DP alignment method achieves reasonably good alignment performance (sometimes comparable to its non-private counterpart) with relatively low privacy costs on standard LLMs and benchmark datasets.

**Strengths:**

- The paper is clearly presented and easy to follow.
- The proposed approach is intuitive and straightforward to implement.
- Experimental results demonstrate promising outcomes, suggesting that this DP alignment method could be readily applied in practice.

**Weaknesses:**

- The approach is a straightforward application of DP mean computation for the alignment vector, which limits the technical contribution of this work.
- Although the topic is novel and could be a promising area for further investigation, the current motivation lacks clarity, particularly around why alignment should be considered a privacy concern. This is especially relevant given that steering could largely be achieved with public (non-sensitive) data, and (non-DP) mean computation is generally regarded as a low-risk operation (as also evidenced in the experimental results).
- The experiments, while indicating potential, currently feel more like proof-of-concept tests rather than a comprehensive study that would meet the standards of a top-tier conference.

**Questions:**

- While rigorous privacy guarantees are generally valuable, the authors could better motivate the topic or provide more natural application scenarios for DP alignment. As noted, (1) alignment could largely be achieved with public data, (2) non-DP linear alignment generally poses low privacy risk, as the "mean" operation aggregates population-level information, reducing individual influence, and (3) the threat model could be clarified further, with comparisons of attack performance under various scenarios to highlight potential privacy threats.

- The description in Section 6 could be more detailed. Adding a formal presentation, diagram, or pseudocode for the attack would improve clarity and self-containment. Additionally, more guidance on selecting canaries and evaluating sensitivity with different choices would be beneficial.

- To enhance the study’s completeness, the experiments could include sensitivity analysis to different clipping bounds (a key factor in practical DP applications), performance across varying epsilon/noise scales, results on diverse datasets across different modalities or semantics, and comparisons of alignment performance on public vs. private datasets (considering potential distribution shifts). Including results with language models of varying structures would also provide a broader perspective.

- Could the authors clarify how PCA steering is conducted (e.g., using labels in a supervised manner)? The notably low performance of PCA steering compared to other methods is somewhat surprising and warrants further explanation.

- In Figure 3, the "zero-shot" reference line appears to be missing for "AI Coordination" on "Llama-2 7B."

- The placement of qualitative examples for Sections C.6 and C.7 in the Appendix could be adjusted to improve readability.

---

> ### Author Response · Authors · 2024-11-20
> **Official Response by Authors (Part 1)**
>
> We thank the reviewer for their careful review and raising concerns and comments. We address each of them below. To answer some of the queries, we have also reported new experimental evaluations in particular: a) results with two additional LLMs (Mistral and Gemma), b) ablation to study the impact of i) noise and ii) clipping factor.
>
>
> > Q. **“why alignment is a privacy concern? ”**
>
> A. We thank the reviewer for raising this important point. We believe LLM alignment is an emerging research area which deserves careful privacy considerations.
> 1. Multiple recent works have highlighted privacy concerns of LLM alignment data. In particular, Feng et. al. [1] showed how to conduct Membership Inference Attacks on DPO and PPO-based alignment for LLMs. In another recent work, Seyitoğlu et. al. [7] showed how to extract memorized data from unlearned LLMs with steering vectors. These works suggest that this is an emerging area of research interest which our work aims to provide a solution for.
> 2. Apart from the interesting research question, several real-world situations require the alignment data to be private.
>     1. In L079-086 we introduce a real-world scenario which can potentially pose privacy risks in deployed LLMs - A hospital using patient history to align LLMs via steering for generating contextually relevant responses (precision medicine). Such healthcare data must necessarily remain private unless special consent is given.
>      2. Industries spend huge resources to collect expensive, high-quality hand-curated alignment data which is generally not released publicly [10,11]. Being able to maintain their private control over this data while aligning their LLMs, will lead to wider adoption of LLM alignment algorithms.
> 3. Besides LLM alignment, steering methods are also increasingly useful for LLM instruction-tuning [2], safety[4], refusal [3], jailbreaks[6], backdoor triggers [5],  and unlearning[7]. Private data can be used in any of these diverse applications which warrants a careful look at the privacy implications of steering algorithms, hence the focus of our current work.
>
> We will include this discussion in the revised draft.
>
> > Q. **The experiments, while indicating potential, currently feel more like proof-of-concept tests rather than a comprehensive study that would meet the standards of a top-tier conference.**
>
> A. We respectfully disagree with the assessment that our work is a proof-of-concept test and not a comprehensive study. In particular, Reviewers R52E and DS5g have acknowledged our comprehensive experiments conducted across seven benchmark datasets in our study. We have provided results on seven benchmarks exhibiting real-world behaviors, four model families of state of the art open-source LLMs, on i) alignment evaluation ii) MMLU iii) scaling across model size . This is a significantly comprehensive empirical study, containing more experimental evaluations than other works in this field. Nevertheless, we have provided more ablations and sensitivity analysis in the rebuttal which we hope will address the reviewer’s concerns.
>
> > Q. **“straightforward application of DP, limited technical contribution”**
>
> A. While we agree that the algorithm is simple, we believe this is a strength of our work and not a weakness. Similar simple solutions have proven effective in related domains. For example, the original non-private steering vector, which is a relatively simple application of averaging representations and adding it to the trained network, usually outperforms more complex techniques like PCA-based steering vector..
>
> In this particular case of private steering, we would like to highlight that
> 1. Our approach already delivers strong and consistent results across four model families and seven benchmark datasets. Hence we believe it is not always helpful to develop complex algorithms when a simple solution does not already exist in literature but performs well (as our experiments show).
> 2. Simpler algorithms also have other benefits. Our approach is easy to implement and is more likely to be adopted in practical and real-world scenarios. It is also easier to analyse and more likely to be free of bugs.
>
> Overall, we do not think this is a weakness of the paper but rather a strength that such a simple application of DP works consistently well across a wide variety of benchmarks.
>
>
>
> > Q. **“zero-shot” line missing in one case**
>
> A. We thank the reviewer for pointing this out. This was an oversight and has been corrected in the updated draft. The zero-shot results for AI Coordination with Llama-2 and Qwen-2.5 are 24.2 and 9.5, respectively.

---

> ### Author Response · Authors · 2024-11-20
> **Official Response by Authors (Part 2)**
>
> > Q. **“more experiments - different clipping bounds, varying noise levels, different LLMs”**
>
> A. As suggested by the reviewer, we have conducted the following experiments to strengthen our empirical findings. The additional experiments have also been added to the paper in Figure 3. New results of the ablations are added in Figure 5 and Appendix D.1
>
> 1. **Results with language models of varying structures**: We report the results comparing PSA and non-private Mean steer across all benchmark datasets with two more models belonging to diverse model families - Mistral and Gemma. We already report the results with Llama and Qwen in Section 5.2.1.
>
> |  Model      |            | Sycophancy | Hallucination | Refusal | Survival Instinct | Myopic Reward | AI Coordination | Corrigibility |
> |---------|------------|:----------:|:-------------:|:-------:|:-----------------:|:-------------:|:---------------:|:-------------:|
> | Mistral | Mean Steer |    78.4    |      75.9     |   94.5  |        72.9       |      63.7     |       52.6      |     92.05     |
> |         |     PSA    |    76.4    |      70.7     |   87.6  |        70.4       |       59      |       56.5      |      80.5     |
> |  Gemma  | Mean Steer |    53.8    |     55.01     |   41.8  |        52.1       |      47.3     |       54.2      |      55.3     |
> |         |     PSA    |     53     |      54.6     |   40.6  |        50.9       |      47.1     |       53.1      |      55.2     |
>
> Our findings with **with Mistral and Gemma also show low utility degradation due to PSA** in line with our observations in Figure 3.
>
> 2. **Performance across varying noise scales**: Here, we report the results across all benchmark datasets with Llama-2 7B to investigate the impact of increasing noise on the alignment performance.
>
> | Noise     | Sycophancy | Hallucination | Refusal | Survival Instinct | Myopic Reward | AI Coordination | Corrigibility |
> |------|------------|---------------|---------|-------------------|---------------|-----------------|---------------|
> | 0.02 |      68.09 |          84.9 |    73.4 |              49.2 |          79.8 |           29.26 |          88.4 |
> | 0.04 |       66.4 |          83.6 |      73 |              46.5 |          79.5 |            33.1 |         86.02 |
> | 0.06 |       61.3 |          82.2 |    71.8 |              43.4 |          75.8 |            32.9 |          85.4 |
> | 0.08 |       60.3 |          80.1 |      70 |                43 |            75 |            28.4 |          81.4 |
>
> As expected, **increasing noise leads to larger drops in utility**.
>
> 3. **Sensitivity analysis to different clipping bounds**: Here, we report the impact of increasing the clipping threshold on the performance of Llama-2 7B across all benchmark datasets.
>
> |  Clipping Threshold  | Sycophancy | Hallucination | Refusal | Survival Instinct | Myopic Reward | AI Coordination | Corrigibility |
> |----|------------|---------------|---------|-------------------|---------------|-----------------|---------------|
> | 10 |      66.08 |          84.7 |    72.5 |              48.5 |          82.6 |            35.8 |          91.5 |
> | 15 |       66.9 |          83.8 |    71.3 |             51.06 |          81.4 |            35.7 |          89.8 |
> | 20 |       66.1 |          82.6 |    70.7 |              48.1 |          80.9 |            34.5 |         89.07 |
> | 25 |       65.7 |            82 |    70.3 |              43.8 |          79.4 |            31.5 |          87.7 |
>
> We observe that **at the same noise level, larger clipping threshold leads to a drop in utility**.  The clipping threshold here is similar to that in DP-SGD and influences utility in two ways listed below, and our experiments show that usually the first effect dominates:
>
> i. **Larger thresholds increase effective noise**: While our algorithm adds the same noise to the model regardless of the threshold, the vectors are divided by the threshold before noise addition. Therefore, a larger threshold effectively reduces the signal-to-noise ratio, thereby decreasing utility.
>
> ii. **Smaller thresholds introduce bias**: When the clipping threshold exceeds the maximum norm of the difference vectors, no clipping occurs, preserving the original distribution of the vectors and leads to an unbiased estimator. In contrast, when the clipping threshold is small, only the larger vectors are clipped, altering the distribution and introducing bias into the mean estimator, which also decreases utility.

---

> ### Author Response · Authors · 2024-11-20
> **Official Response by Authors (Part 3)**
>
> > Q. **“More descriptive Section 6. Need info on canary selection”**
>
> A. We have added Section D.2 to describe the MIA in more detail. Here we provide a brief description.
> 1. **Canary Selection**: We create a set $\mathcal{S}$ of canary words that are gibberish yet plausible, each 6-7 letters long and starting with the same letter. This approach follows prior work [14,15] and uses realistic nonce words \citep{malkin2021gpt} to simulate practical data leakage.
> 2. **Canary Pairing**: From $\mathcal{S}$, we randomly select three canaries: one anchor $a$ and two targets $t_1$​ and $t_2$​. These form pairs $z_1=(a,t_1)$​ and $z_2=(a,t_2)$.
> 3. **Attack Procedure**: In each MIA trial, we randomly insert either $z_1$ or $z_2$ into the dataset alongside benign samples to create the steering vector (let’s say $z_1$ is inserted). We then prompt the LLM $N$ times with the anchor $a$ and count how often the target $t_1$ appears in the outputs. If the count exceeds a threshold $\tau$, we infer that $z_1$​ was used, indicating membership.
>
>
> > Q. **“clarify PCA steering**
>
> A. We follow the algorithm introduced in [9] as the PCA setting. In this setting, the last token activations of the negative and positive demonstrations are computed and their difference vector is stored. The difference vector approach is used in existing steering works like [3,4,5,6]. Instead of averaging the difference vector over all samples (which is the case in mean steering), **PCA steering computes the top principal direction (PCA with k=1) of the difference matrix. Then this top principal component is used as the steering vector.** For more details see Section 3 in [9]. If the reviewer prefers, we will include this pseudocode in the appendix of our paper as well.
>
> [1] Feng, Qizhang, et al. "Exposing privacy gaps: Membership inference attack on preference data for llm alignment." arXiv preprint arXiv:2407.06443 (2024).
>
> [2] Stolfo, Alessandro, et al. "Improving Instruction-Following in Language Models through Activation Steering." arXiv preprint arXiv:2410.12877 (2024).
>
> [3] Arditi, Andy, et al. "Refusal in language models is mediated by a single direction." arXiv preprint arXiv:2406.11717 (2024).
>
> [4] Cao, Zouying, Yifei Yang, and Hai Zhao. "Nothing in excess: Mitigating the exaggerated safety for llms via safety-conscious activation steering." arXiv preprint arXiv:2408.11491 (2024).
>
> [5] Price, Sara, et al. "Future events as backdoor triggers: Investigating temporal vulnerabilities in llms." arXiv preprint arXiv:2407.04108 (2024).
>
> [6] Ball, Sarah, Frauke Kreuter, and Nina Panickssery. "Understanding jailbreak success: A study of latent space dynamics in large language models." arXiv preprint arXiv:2406.09289 (2024).
>
> [7] Seyitoğlu, Atakan, et al. "Extracting Unlearned Information from LLMs with Activation Steering." arXiv preprint arXiv:2411.02631 (2024).
>
> [8] Parikh, Rahil, Christophe Dupuy, and Rahul Gupta. "Canary extraction in natural language understanding models." ACL 2022
>
> [9] Liu, Sheng, et al. "In-context vectors: Making in context learning more effective and controllable through latent space steering." ICML 2024
>
> [10] Dubey, Abhimanyu, et al. "The llama 3 herd of models." arXiv preprint arXiv:2407.21783 (2024).
>
> [11] Yang, An, et al. "Qwen2 technical report." arXiv preprint arXiv:2407.10671 (2024).
>
> [12] Carlini, Nicholas, et al. "Extracting training data from large language models." 30th USENIX Security Symposium (USENIX Security 21). 2021.
>
> [13] Carlini, Nicholas, et al. "The secret sharer: Evaluating and testing unintended memorization in neural networks." 28th USENIX security symposium (USENIX security 19). 2019.
>
> [14] Zanella-Béguelin, Santiago, et al. "Analyzing information leakage of updates to natural language models." ACM CCS 2020
>
> [15] Millière, Raphaël. "Adversarial attacks on image generation with made-up words." arXiv preprint arXiv:2208.04135 (2022).

---

> ### Author Response · Authors · 2024-11-25
> **Gentle Reminder**
>
> Dear reviewer L9b9, we deeply appreciate the time and effort you invested in reviewing our paper. Since the discussion period closes soon, we hope our response and additional experiments address all your concerns appropriately and that you will reconsider the score accordingly.
>
> We look forward to further discussion on any questions or comments you still have. Please let us know.

---

> > ### Comment · Reviewer_L9b9 · 2024-11-25
> >
> > Thank you for the rebuttal, which has addressed my main concerns. As a result, I have increased my score.

---

### Official Review · Reviewer_R52E · 2024-11-07

**Soundness:** 3
**Presentation:** 3
**Contribution:** 3
**Rating:** 6
**Confidence:** 4

**Summary:**

This paper studies the private streering for large language models alignment via activation editing. Based on previous work that (update to complete the summary, no change for other comments) computing the activations of several layers could align model with human values, this work propose PSA to privatize the average of activations of each layers to protect the sensitive data with differential privacy. The evaluations include multi-choice and open-ended generations tasks ranging the alignment performance and the general capabilities. This work also did empirical privacy leakage estimation for the proposed method.

**Strengths:**

The experiments are thorough, measuring the alignment performance, text generation quality, general capability, and designing privacy evaluation.

**Weaknesses:**

1. One non-private baseline is missing. Wu et al. 2024 provides the non-private baseline both for setting $\sigma=0$ for their methods and original non-private method. It seems to me that this work only provides the non-private one while the $\sigma=0$ is missing. It would be helpful to understand the clipping impact. For example, it seems to me that the norm of the non-private activations steering is unknown and maybe it shows different distribution than the norms are all within 1. If so, it is unclear to me why the private version and non-private version has similar performance.

2. The technical contribution is limited. The non-private version is to compute the activation and average, and the private method proposed in this work is to privatized such mean value. I appreciate the experimental evaluation, and I think it would be helpful to provide the codebase to improve reproducibility. Besides, I wonder if there is any reference for the mean steering and other advanced steering (no new experiments for this result needed, more like a reference). It would be also better to provide the privacy parameters $\sigma$ to improve reproducibility.  For example, when stating $\delta=1/(5n)$, I wonder if it is $\delta=1/(5n)$ for the full model? If so, then each layer should allocate $\delta=1/(25n)$ with 5 layers as editing. Similarly, it is unclear the $\varepsilon=2$ in Table 2 is for the all five layers or just a single layer. Also, the privacy estimation seems to be very tight for PSA ($\varepsilon=1.6$ vs $\varepsilon=2$), while prior work like Nasr et al. 2021 needs to design gradient canary rather than input canary and train more than 10k models to achieve such tight analysis. As empirical privacy estimation only provides the the lowest $\varepsilon$, it would be helpful to provide the privacy parameters to understand the canary property in this case.

I am willing to increase my score if my concerns addressed.

**Questions:**

See above.

---

> ### Author Response · Authors · 2024-11-17
> **Clarification on Reference for Wu et al. 2024**
>
> We thank the reviewer for their thoughtful feedback on our submission. Unfortunately, we were unable to locate the specific reference for Wu et al. 2024. Could you kindly provide additional details, such as a title, or a link? This would help us better address your feedback in our rebuttal. We will be happy to report additional results during our discussion.

---

> > ### Comment · Reviewer_R52E · 2024-11-18
> >
> > Wu et al. 2024a. considers the two non-private baseline $\varepsilon=\infty$. As in their Table 1, setting $\sigma=0$ for their algorithms and the original ICL pipeline. For the initial original w1, I was wondering the effect of clipping of PSA.
> >
> > Tong Wu, Ashwinee Panda, Jiachen T Wang, and Prateek Mittal. Privacy-preserving in-context learning for large language models. In International Conference on Learning Representations (ICLR), 2024a.

---

> ### Author Response · Authors · 2024-11-20
> **Official Response by Authors (Part 1)**
>
> We thank the reviewer for their careful review and the two sets of specific questions. We are also encouraged by the reviewer's scores on Soundness, Presentation, and Contribution. We hope the answers below, along with the new experimental data, satisfactorily answer the reviewer’s queries, and they will consider increasing their scores.
>
> > Q1. **“One non-private baseline is missing. …. It would be helpful to understand the clipping impact. ”**
>
> A. As we understand it, the reviewer is asking how the clipping affects the whole algorithmic pipeline independent of the added noise. We thank the reviewer for this interesting question as clipping indeed plays an important role here in both i) improving the utility when no noise is added and ii) allowing for a smaller epsilon by controlling the sensitivity as in classical additive noise mechanisms. To illustrate this, we a) run PSA with different clipping thresholds for $\sigma=0$ as requested by the reviewer and b) provide intuition on why less restrictive clipping is useful as well as cite related works with similar evidence.
>
> 1. **Impact of clipping when $\sigma=0$**: Here, we provide the results with Llama-2 7B across all benchmark datasets in the noiseless setting  (sigma = 0). **This is a similar setting as the one reported in Wu et al. 2024**.
>
> | Clip Threshold | Sycophancy | Hallucination | Refusal | Survival Instinct | Myopic Reward | AI Coordination | Corrigibility |
> |:--------------:|------------|---------------|---------|-------------------|---------------|-----------------|---------------|
> |       10       |       67.1 |         85.98 |    74.2 |              48.3 |          82.5 |            36.2 |          91.4 |
> |       15       |       66.8 |          85.4 |    73.8 |              47.8 |          81.7 |            36.1 |          90.3 |
> |       20       |       66.1 |          84.7 |    73.5 |              47.2 |          81.2 |            35.9 |          89.6 |
> |       25       |       65.5 |          83.2 |    72.8 |              46.1 |          80.8 |            32.1 |          88.1 |
>
> We observe that **increasing the clipping threshold leads to a decrease in utility**.
>
> 2. **Intuition for why clipping improves performance** Here, we provide some intuitive explanations for why this might be the case. Clipping and Normalisation (L227 in Algorithm 1)  involve two steps: first, the difference vector is clipped to $C_l$, using $d_i^l = d_i^l / \max(1, \frac{||d_i^l||_2}{C_l})$; then, the clipped difference vector is divided by $C_l$ to ensure that $\bar{d}_i^l$ has norm smaller than 1.
>     1. **Step 1: smaller thresholds introduce bias**: When the clipping threshold is larger than the maximum norm of the difference vectors, no clipping occurs, preserving the original distribution and keeping the estimator unbiased. If the threshold is small, larger vectors are clipped, altering the distribution and introducing bias into the mean estimator, which reduces utility.
>     2. **Step 2: threshold controls steering vector influence**: In a noiseless setting, the final model parameter is the original parameter plus $\frac{1}{C_l}$ times the averaged clipped vectors. The threshold $C_l$ controls the influence of the steering vector. A larger $C_l$ reduces the steering vector's impact, which can either improve or decrease utility.
>
> Based on our experiments, we observe that larger clipping thresholds lead to utility degradation in general, showing that the normalization step dominates. We have added this experiment to the paper (Section E.1) as part of the ablation study.
>
> 3. **Past work on Clipping for non-private training:** Previous work [7] shows that normalizing LLM representations to unit norm improves performance, increases embedding space separability, and speeds up training convergence. Figure 4 in [7] demonstrates that normalized transformer representations are better conditioned, leading to better convergence. Although this differs from our setting, it supports the idea that normalized representations improve performance in the LLM setting. The above experiments (clipping with $\sigma=0$) show similar benefits for non-private steering. To the best of our knowledge, our work is the first to show this benefit for normalized against unnormalized steering vectors, which could be explored in future research.
>
>
> > Q. **“References for mean steering and other steering methods”**
>
> A. In L120-121, we provide references for several steering methods in the literature [1,2]. More recent and contemporaneous works that use steering methods include [3,4,5,6].
>
> > Q. **“provide the codebase to improve reproducibility”**
>
> A. We agree and acknowledge the importance of reproducibility of results. We plan to publicly release all code, datasets and private steering vectors used in this study.

---

> ### Author Response · Authors · 2024-11-20
> **Official Response by Authors (Part 2)**
>
> > Q. **“It would also be better to provide the privacy parameters \sigma to improve reproducibility… It’s unclear whether the $\epsilon=2$ in Table 2 is for all five layers or just a single layer.”**
>
> A. In our experiments, we set $\sigma = 0.02$ and provide the epsilon values in Table 2, and the computation of per-layer $\epsilon$ in terms of $\sigma$ and per-layer $\delta$ is presented in L233 of Algorithm1. We set per-layer $\delta$ to be $\frac{1}{5n}$, which results in an overall $\delta = \frac{1}{n}$. We have added Table 17 and Figure 5 in the updated draft to illustrate the effect of sigma on the LLM performance.
>
>
> Also, we modified Table 2 to include the per-layer epsilon as well as the total epsilon for each dataset.
> | Attribute            | Sycophancy | Hallucination | Refusal | Survival Instinct | Myopic Reward | AI Coordination | Corrigibility |
> |----------------------|------------|---------------|---------|-------------------|---------------|-----------------|---------------|
> | Per-layer $\epsilon$ | 0.4        | 0.4           | 0.94    | 0.46              | 0.42          | 1.08            | 1.32          |
> | Total $\epsilon$     | 2.0        | 2.0           | 4.7     | 2.3               | 2.1           | 5.4             | 6.6           |
>
> > Q. **“tight privacy estimation. Nasr et al 2021 had to design gradient canary and train 10k models to achieve such tight analysis”**
>
> A. The privacy preserving mechanisms audited by Nasr et. al. (DP-SGD) and us (PSA) are very different and thus they require different auditing algorithms. Our algorithm involves a single noise addition to a one-step optimisation process (mean), in contrast to the multiple randomized iterations (shuffled stochastic batch gradient descent) in DP-SGD. Moreover, the steering vector in our case is typically computed using a smaller dataset, as compared to the size of training sets for DP-SGD. These factors potentially make the steering vector more susceptible to memorization than DP-SGD. Other recent work [8] has also provided evidence of a successful memorized data extraction attack for steering vectors.
>
> [1] Liu, Sheng, et al. "In-context vectors: Making in context learning more effective and controllable through latent space steering." ICML 2024
>
> [2] Panickssery, Nina, et al. "Steering llama 2 via contrastive activation addition." ACL 2024
>
> [3] Qiu, Yifu, et al. "Spectral Editing of Activations for Large Language Model Alignment." NeurIPS 2024
>
> [4] Wang, Haoran, and Kai Shu. "Backdoor activation attack: Attack large language models using activation steering for safety-alignment." CIKM 2024
>
> [5] Cao, Zouying, Yifei Yang, and Hai Zhao. "Nothing in excess: Mitigating the exaggerated safety for llms via safety-conscious activation steering." arXiv preprint arXiv:2408.11491 (2024).
>
> [6] Cao, Yuanpu, et al. "Personalized Steering of Large Language Models: Versatile Steering Vectors Through Bi-directional Preference Optimization." arXiv preprint arXiv:2406.00045 (2024).
>
> [7] Loshchilov, Ilya, et al. "ngpt: Normalized transformer with representation learning on the hypersphere." arXiv preprint arXiv:2410.0113 1 (2024).
>
> [8] Seyitoğlu et al. “Extracting Unlearned Information from LLMs with Activation Steering.” arXiv preprint arXiv:2411.02631 (2024).

---

> > ### Comment · Reviewer_R52E · 2024-11-25
> >
> > Thank you for your detailed response. I find my concerns solved and therefore increasing my score to 6.

---

### Author Response · Authors · 2024-11-20
**General Response by Authors**

We sincerely thank all reviewers for their detailed feedback and insights regarding our paper. We are pleased to see that the reviewers appreciated several aspects of our work including:

1. **Importance of alignment privacy**: Both Reviewer DS5g and Hvwa emphasize the significance of our focus on privacy auditing during LLM alignment, recognizing its importance.
2. **Thorough experiments**: Reviewers R52E and DS5g acknowledge the comprehensive experiments conducted across seven benchmark datasets in our study.
3. **New MIA for steering**: Reviewers DS5g and R52E specifically appreciated our novel Membership Inference Attack for LLM steering.
4. **Paper Writing**: Reviewers Hvwa and L9b9 noted the clarity and writing in our paper.


We have carefully considered the feedback provided and revised the paper accordingly to address the concerns raised by the reviewers:
- We conduct additional experiments with results from **two more LLMs (Mistral and Gemma)** across all seven benchmarks.
- We conduct ablation studies to investigate the **impact of noise and clipping factors** on LLM performance.
- We explain the **MIA in more detail.**

We hope that the individual responses and revisions adequately address the reviewers’ concerns and clarify the points raised. We look forward to further feedback and discussion on our contribution. We appreciate the effort each reviewer has taken to evaluate our work.

---

### Meta-Review · Area_Chair_NDDB · 2024-12-21

**Metareview:**

The submission proposes Private Steering for LLM Alignment (PSA), an algorithm that performs activation editing with a differential privacy (DP) guarantee on a sensitive dataset. The submission empirically evaluates the performance of this algorithm, comparing it to non-DP activation editing algorithms as well as the zero-shot capabilities of 3 state-of-the-art LLMs of different sizes on standard alignment benchmarks. The submission also proposes a membership inference attack (MIA) for this setting, and shows that this attack can be successful when non-DP steering is performed.

Overall, the submission studies a timely problem, proposes a simple DP algorithm, and performs a thorough evaluation of its performance.

**Additional Comments On Reviewer Discussion:**

The reviewers raised different concerns including:

1) Limited technical contribution, as the DP algorithm is based on a straightforward application of the standard Gaussian mechanism from the DP literature
2) Some regimes of parameters were missing from the evaluation
3) The motivation for studying privacy in alignment
4) The experimental evaluation

In the discussion, the authors satisfactorily address concerns 2), 3) and 4). While concern 1) mostly remains, the reviewers are in agreement that the pros of the paper outweigh its cons.

---

### Decision · Program_Chairs · 2025-01-22

Accept (Poster)